

# The energetic effect of hip flexion and retraction in walking at different speeds: a modeling study

Jian Jin[1], Dinant Kistemaker[1], Jaap H. van Dieën[1], Andreas Daffertshofer[1,2] and Sjoerd M. Bruijn[1,2]

[1] Department of Human Movement Sciences, Faculty of Behavioural and Movement Sciences, Vrije Universiteit Amsterdam, Amsterdam, The Netherlands
[2] Institute of Brain and Behavior Amsterdam, Amsterdam, The Netherlands

## ABSTRACT

In human walking, power for propulsion is generated primarily *via* ankle and hip muscles. The addition of a 'passive' hip spring to simple bipedal models appears more efficient than using only push-off impulse, at least, when hip spring associated energetic costs are not considered. Hip flexion and retraction torques, however, are not 'free', as they are produced by muscles demanding metabolic energy. Studies evaluating the inclusion of hip actuation costs, especially during the swing phase, and the hip actuation's energetic benefits are few and far between. It is also unknown whether these possible benefits/effects may depend on speed. We simulated a planar flat-feet model walking stably over a range of speeds. We asked whether the addition of independent hip flexion and retraction remains energetically beneficial when considering work-based metabolic cost of transport (MCOT) with different efficiencies of doing positive and negative work. We found asymmetric hip actuation can reduce the estimated MCOT relative to ankle actuation by up to 6%, but only at medium speeds. The corresponding optimal strategy is zero hip flexion and some hip retraction actuation. The reason for this reduced MCOT is that the decrease in collision loss is larger than the associated increase in hip negative work. This leads to a reduction in total positive mechanical work, which results in an overall lower MCOT. Our study shows how ankle actuation, hip flexion, and retraction actuation can be coordinated to reduce MCOT.

## INTRODUCTION

When humans walk they seemingly minimize the energy spent per distance travelled, *i.e.*, the metabolic cost of transport (MCOT) (*Bertram & Ruina, 2001*; *Ralston, 1958*). Ankle plantar flexors, hip flexors and hip extensors provide the bulk of the required energy (*Farris & Sawicki, 2012*; *Montgomery & Grabowski, 2018*; *Winter, 1983*). Impairment of ankle plantar flexors or hip muscles may result in gait adaptations and reduced energy efficiency (*Farris et al., 2015*; *Huang et al., 2015*; *Stevens, Podeszwa & Tulchin-Francis, 2019*). Such changes may also happen due to aging (*Franz, 2016*; *McGibbon, 2003*).

Corresponding author
Sjoerd M. Bruijn, s.m.bruijn@vu.nl

To unravel the mechanisms of such gait adaptations, it is essential to understand the energetic effects of ankle and hip actuation. The pioneering work of *Kuo (2002)* revealed that impulsive push-off applied *via* the trailing leg just before heel strike can substantially reduce the energy losses associated with collision of the leading leg. It can thus reduce the total mechanical work required to maintain periodic gaits. In Kuo's model, however, toe-off impulse was applied at the (point) foot, to represent push-off forces of the stance leg generated by muscles at the ankle, knee and hip joints (*Kuo, 2002*).

Ankle and hip actuation have different functional roles during walking. Ankle actuation mainly contributes to center-of-mass acceleration during push-off and swing leg initiation before toe-off (*Zelik & Adamczyk, 2016*). Hip actuation mainly plays three roles: (i) push-off *via* the hip extension torque during the late stance phase (*DeVita & Hortobagyi, 2000*; *Winter, 1983*); (ii) weight acceptance during the first half of stance phase (*Winter, 1980*); and (iii) acceleration and deceleration of the swing leg during the early and late swing phase respectively (*Doke, Donelan & Kuo, 2005*; *Muybridge, 2012*). The last role enables hip muscles to modulate step length and frequency, which is the focus of our study.

Several modeling studies demonstrated energetic benefits of spring-like elasticity around ankle and hip (*Bregman et al., 2011*; *Duindam, 2006*; *Hasaneini, 2014*; *Kerimoglu et al., 2021*; *Kuo, 2002*; *O'Connor, 2009*; *Zelik et al., 2014*). The addition of a torsional hip spring can reduce collision loss by reducing the step length, thus requiring less positive mechanical work in a periodic gait (*Kuo, 2002*). When hip actuation is considered a conservative spring with zero net mechanical work over a gait cycle, its associated metabolic cost is often ignored (*Kuo, 2002*; *Zelik et al., 2014*). As already noted by *Kuo (2002)* and *Zelik et al. (2014)*, generating hip torques is not for free: hip torques demand metabolic energy due to muscle (de-)activation and cross-bridge cycling (*Homsher & Kean, 1978*; *Woledge, Curtin & Homsher, 1985*). This raises the question whether a reduced collision loss outweighs the increase in metabolic cost due to hip actuation. *Hasaneini et al. (2013)* optimized the work-based metabolic cost of a model with telescoping ankle push-off and hip actuation during the swing and stance phase. They found an optimal gait with ankle push-off and hip actuation contributing equally to the metabolic cost. Why this actuation strategy is most energy efficient, however, has not been answered. *Kuo (2001)* tested whether minimizing the metabolic energy in simple walking models pinpoints the relationship between walking speed and step length in humans. When modeling the metabolic cost of the push-off impulse as proportional to its mechanical work, and when modeling the metabolic cost of the spring-like hip torque as proportional to its (peak) force rate, it does. Yet, *Kuo (2001)* noted that the work done by spring-like hip torque was not included in the metabolic cost. If account for, it may have resulted in burst-like hip impulses as observed in humans (*Doke, Donelan & Kuo, 2005*).

Adding hip actuation can be beneficial in terms of reducing the MCOT (*Hasaneini et al., 2013*; *Kuo, 2001*; *Kuo, 2002*; *Zelik et al., 2014*). It is also essential for achieving high walking speeds (*Dean & Kuo, 2009*). Whether and how hip actuation, when modelled as independent hip flexion and retraction actuation, can reduce the MCOT compared to ankle actuation only for a variety of walking speeds, is an open question. To address this, we investigated the effect of hip flexion and retraction actuation on the MCOT at various

speeds. The model we used is a planar flat-feet walker model actuated by ankle and hip torques. The ankle actuation was similar to *Zelik et al. (2014)*, which enables a non-instantaneous ankle push-off or double stance phase. Our hip flexion and retraction actuation were modelled by two independent hip springs switching on before and after zero-crossing of the hip angle, respectively. We compared the MCOT of stable periodic gaits with varying hip flexion and retraction actuation at each speed, then identified and analyzed the optimal actuation strategy in terms of the lowest MCOT. We further analyzed how mechanical and metabolic energy components were influenced by different levels of hip flexion and retraction actuation.

## METHODS

We employed a simple planar flat-feet walker model to investigate the effect of hip actuation on the estimated MCOT over a range of walking speeds. The amount of hip flexion and retraction actuation was varied at each speed, and the effects on the estimated MCOT were studied. We identified stable periodic gaits at each speed that minimized the estimated MCOT, and analyzed why these gaits are optimal.

### Model

Our model consisted of four rigid segments (with inertia) representing two straight legs and two flat feet, connected in three frictionless hinge joints, one representing both hips and two representing the ankle joints. The model's geometric and mass distribution parameters were almost similar to Kuo's anthropomorphic walker (*Kuo, 2002*), except that we replaced the circular feet with flat feet (length 0.15 and center-of-mass located at the foot midpoint). The ankle joint in our model allowed for a non-instantaneous double stance phase; see Fig. 1; see Table 1 for parameter settings. To account for variations in body mass and limb morphology, the total mass and leg length were used to normalize the model parameters: masses are given as a proportion of the total mass $m^{tot}$, time was rescaled by $\sqrt{l/g}$, speed by $\sqrt{gl}$, ankle and hip spring stiffness by $m^{tot}g/l$, ankle damping coefficient by $m^{tot}\sqrt{g/l}$, work by $m^{tot}gl$, with $g$ denoting the gravitational constant. As a result, a dimensionless value of 1 of speed and step length corresponds to a speed of 3.1 m/s and a step length of 1 m. From here-on we will consider all configuration parameters in normalized units unless specified otherwise. We defined generalized coordinates $q = \begin{bmatrix} \phi_1 & \cdots & \phi_4 & \dot{\phi}_1 & \cdots & \dot{\phi}_4 \end{bmatrix}^T$, where each $\phi_i$ is the angle between the positive $x$-axis and the line from the proximal to the distal joint. The subscripts refer to the segments numbered along a kinematic chain starting with the toe of the stance foot (*i.e.*, 1 = stance foot, 2 = stance leg, 3 = swing leg, 4 = swing foot). Each segment's configuration was determined by four parameters: center-of-mass $m_i$, length $l_i$, distance of the segment's center-of-mass to the segment's distal end $d_i$, and moment of inertia relative to the center-of-mass $j_i$. With this definition, $d_2 + d_3$ equals leg length $l$; see Fig. 1 and cf. Table 1 for the geometric and mass distribution parameters. The joint torques from proximal to distal were defined as $\begin{bmatrix} T_1^a & T_2^h & T_3^a \end{bmatrix}^T$ where 'a' and 'h' indicate ankle and hip, respectively. The dynamics of the model was formulated as a four-segment chain with the stance foot/

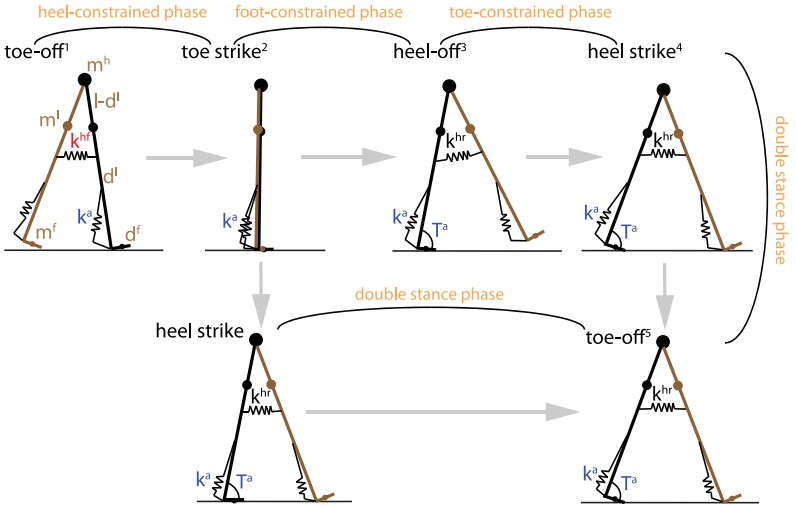

**Figure 1 The planar flat-feet walker and its gait cycle.** The walker may or may not have a toe-constrained phase, depending on whether the heel leaves the ground before contralateral heel strike. For ankle actuation, the walker is implemented with an ankle spring $k^a$ and pulse torque $T^a$ which is activated from the peak ankle dorsiflexion until the end of the gait cycle (toe-off). Dampers were added to the ankle joint of the swing leg after toe-off and of the leading foot after heel strike to reduce oscillations. For hip actuation, the walker is implemented with a hip flexion spring $k^{hf}$ and retraction spring $k^{hr}$ activated before and after the hip angle reaches zero.     

**Table 1 The model parameters of the flat-feet walker.**

| Parameter | Symbol | Value in dimensionless units |
|---|---|---|
| Total mass | $m^{tot}$ | 1 |
| Hip and torso mass | $m^h$ | 0.658 |
| Leg mass | $m^l$ | 0.161 |
| Foot mass | $m^f$ | 0.01 |
| Leg length | $l$ | 1 |
| Leg CoM position | $d^l$ | 0.645 |
| Foot length | $l^f$ | 0.15 |
| Foot CoM position | $d^f$ | 0.075 |
| Damper at ankle of the swing leg | $c^a$ | 0.01 |
| Ankle stiffness | $k^a$ | Optimized parameter |
| Ankle pulse torque | $T^a$ | Optimized parameter |
| Hip flexion stiffness | $k^{hf}$ | Optimized parameter |
| Hip retraction stiffness | $k^{hr}$ | Optimized parameter |

heel/toe attached to the ground using the Newton-Euler approach and the framework of *Casius, Bobbert & van Soest (2004)*; see Supplemental Material A for more details.

As illustrated in Fig. 1, a full cycle of the planar feet walker consisted of the following phases: heel-constrained phase, foot-constrained phase, toe-constrained phase and double stance phase, where 'constrained' indicates that the corresponding part of the foot segment contacts the ground. Depending on whether heel-off (preemptive push-off) occurs before

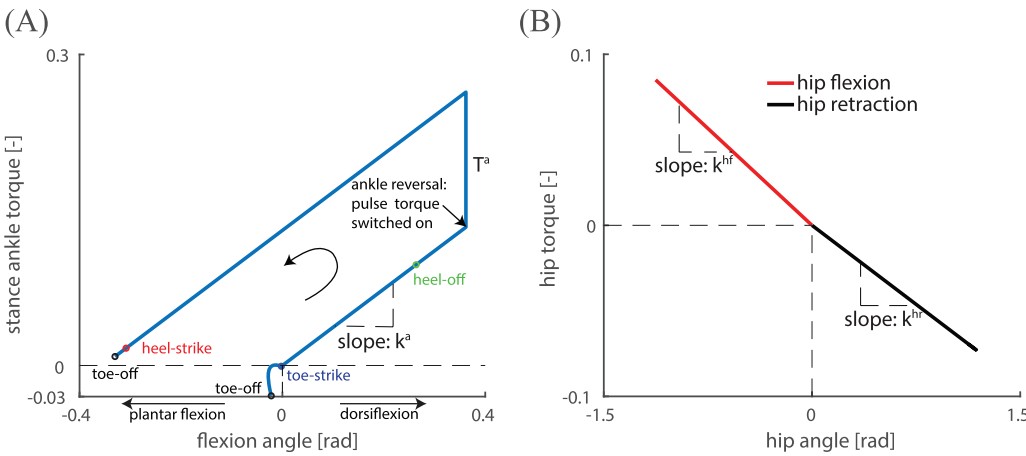

**Figure 2 An illustrative example of the angle-torque relation for (A) ankle actuation and (B) hip actuation with different hip flexion and retraction spring stiffness.** Prior to ankle push-off, elastic energy was stored at the ankle spring. After ankle reversal (from peak ankle flexion until toe-off), the walker was actuated at the ankle by torques generated by the spring in addition to the pulse torque $T^a$. The walker could also be actuated by spring-like hip flexion and retraction torques, which were active only before and after the hip angle reached zero.

or immediately after contralateral heel strike, the toe-constrained phase may or may not take place. As such, a full gait cycle consisted of either three, or four phases (Fig. 1). The transitions between different phases were detected by the following events in temporal order within a gait cycle:

1. Toe-off: vertical ground reaction force on the trailing toe becomes zero. Toe-off moments of the swing foot and of contralateral foot were defined as the beginning and the end of a gait cycle, respectively.

2. Toe strike: stance toe hits the ground, while the heel remains on the ground; the toe strike was assumed to be instantaneous and inelastic (no slip and no bounce).

3. Heel-off: ground reaction force moves to the toe from any other point on the foot; for four-phase gait such an event occurs before heel strike while for three-phase gait it occurs immediately after heel strike.

4. Heel strike: vertical position of the leading heel reaches the ground; in addition, swing leg rotates clockwise ($\dot{\phi}_3 < 0$) to avoid detection of foot-scuffing and to consider only (stable) long-period gaits with swing leg reversal before heel strike (*Kwan & Hubbard, 2007*); the heel strike was assumed to be instantaneous and inelastic.

Similar to *Zelik et al. (2014)*, the walker was actuated at the ankle by torques generated by a spring with stiffness $k^a$ in addition to a constant torque $T^a$ added after ankle reversal (from peak ankle flexion until toe-off) (see Fig. 2A for an illustration). Prior to ankle push-off, elastic energy could be stored by the ankle spring and was subsequently released during ankle push-off (Fig. 2A). To reduce undesirable oscillations, dampers with a small damping coefficient $c^a$ were added to the ankle joint(s) whenever the toe of the corresponding foot segment was not in contact with the ground. For instance, the curve in stance ankle torque from toe-off to contralateral toe strike in Fig. 2A is an effect of this

damper and the spring. The negative work done by dampers during the swing phase was found to be less than 2% of collision loss and was ignored in our analysis. The total ankle torque during push-off (no damper in this phase) is given by:

$$\tau_1^a = T^a + k^a\left(-\frac{\pi}{2} - \phi_2 + \phi_1\right), \tag{1}$$

The total ankle torques outside the push-off phase were defined similarly, but with the constant $T^a$ set to zero.

The walker was also actuated at the hip by flexion and retraction torques, which were active before and after the hip angle reached zero, respectively (see Fig. 2B for an illustration). The hip flexion and retraction torques were used to influence step frequency and step length. The torque-angle relation for hip flexion and retraction was modelled independently by two springs with separate stiffnesses:

$$\tau_2^{hf} = k^{hf}|\pi - \phi_2 + \phi_3|, \tag{2}$$

and

$$\tau_2^{hr} = k^{hr}|\pi - \phi_2 + \phi_3|. \tag{3}$$

The energy losses from inelastic collisions of the heel/toe could be compensated by performing net positive work around the ankle and/or hip joint. This collision loss equaled the opposite of (ground) impulsive work $W_{\text{impulsive}}$, which was computed as half the dot product of impact velocity and (ground reaction) impulse, shown below; for a full derivation, we refer to *Font-Llagunes & Kövecses (2009)*, and for an intuitive proof, see Supplemental Material B.

$$W_{\text{impulsive}} = \frac{1}{2}S_x v_{bx-} + \frac{1}{2}S_y v_{by-}, \tag{4}$$

where $v_{bx-}$, $v_{by-}$ are the horizontal and vertical pre-collision impact velocity of the collision point (*i.e.*, heel), and $S_x$, $S_y$ are the horizontal and vertical ground reaction impulse.

## Numerical simulations and optimizations

We simulated the dynamics of the walker using a Runge-Kutta (4,5) integrator, setting the absolute tolerance and relative tolerance to $10^{-6}$. We found stable periodic gaits by enforcing two conditions: (a) all elements of the end state after walking five steps should be close to initial states $q_0$ within error of $10^{-6}$; (b) the maximum Floquet multiplier (error multiplication factor from step-to-step at post-impact state; see, *e.g.*, *Hurmuzlu & Moskowitz, 1987*; *Wisse & Schwab, 2005*) should be less than 1. We systematically varied speed $v = 0.16{:}0.02{:}0.54$, where 0.54 is the highest speed at which an ankle actuation periodic gait can be found. At each speed, we found optimal gaits with varied hip flexion and retraction actuation: (1) zero hip flexion and retraction actuation (Results section "Energetics of ankle actuation"); (2) hip flexion actuation with zero retraction actuation (Results section "Can MCOT be reduced by adding only hip flexion actuation?"); (3) hip flexion and retraction actuation (Results section "Can adding hip retraction actuation in

addition to hip flexion actuation reduce MCOT?"). In all three cases, the hip actuation parameters $k^{hf}$ and $k^{hr}$ were manipulated or constrained to zero. To minimize the risk of obtaining "local minimum" solutions during the optimization process, we used grid search by sweeping one of the ankle actuation parameters $k^a$ over a feasible range. The remaining *parameter optimization problem* was solved using the Matlab function "fmincon" (with the SQP algorithm), to find the combination of ankle control parameter $T^a$ and eight initial state parameters $q_0$ that minimize the cost function (MCOT) subjected to the constraint of a stable and periodic gait at a given desired speed $v$, given $k^a$, $k^{hf}$ and $k^{hr}$. We used the following cost function as an estimate of the overall 'metabolic' energy required to travel a unit distance (*Schmidt-Nielsen, 1972*).

$$\text{MCOT} = \frac{(\eta^+ W^+ + \eta^- W^-)}{m^{tot} g s} = \frac{(\eta^+ W^+ + \eta^- W^-)}{s}, \tag{5}$$

where $m^{tot}$ is the normalized total mass and $g$ is gravitational constant, which are both 1, $s$ is the step length, $W^+$ is the total positive work from ankle and hip joint, $W^-$ is the total negative work, $\eta^+$ is the inverse of efficiency of generating positive work and $\eta^-$ is the inverse of efficiency of generating negative work. Based on *Margaria (1968)*, we set $\eta^+ = \frac{1}{0.25} = 4$ and $\eta^- = -\frac{1}{1.2} = -0.83$, respectively.

In general, the sum of positive and negative internal work (from ankle and hip) and external work (ground impulsive work, gravity work) equals the change in kinetic energy, which is zero after a periodic gait cycle. Thus, the total negative mechanical work is equal but opposite to the total positive mechanical work from ankle and hip joint. For an example of this relation, see Supplemental Material C, which depicts kinetic energy change and mechanical (internal and external) work performed for a periodic gait within a gait cycle. Note that the negative work performed at collision was not included in the metabolic cost, but collision loss must be compensated by the same amount of positive work performed at ankle and hip, thus the cost of the collision loss is implicitly included in MCOT.

## RESULTS

We compared the MCOT for only ankle actuation, and ankle actuation with hip flexion and/or retraction actuation, when walking at the same speed. We assessed if MCOT could be decreased by adding hip actuation and what the mechanisms were for any difference in MCOT. We focused on a range of speeds at which the model could walk stably and periodically both with and without hip actuation, *i.e.*, 0.16–0.54 in dimensionless units or 0.50–1.69 m/s.

### General model behavior

Before comparing MCOT for these different actuation strategies, we first introduce the basic mechanisms of how the ankle and/or hip actuate the walker by performing (joint) mechanical work, and briefly discuss how metabolic work for ankle/hip actuation is expended in performing mechanical work.

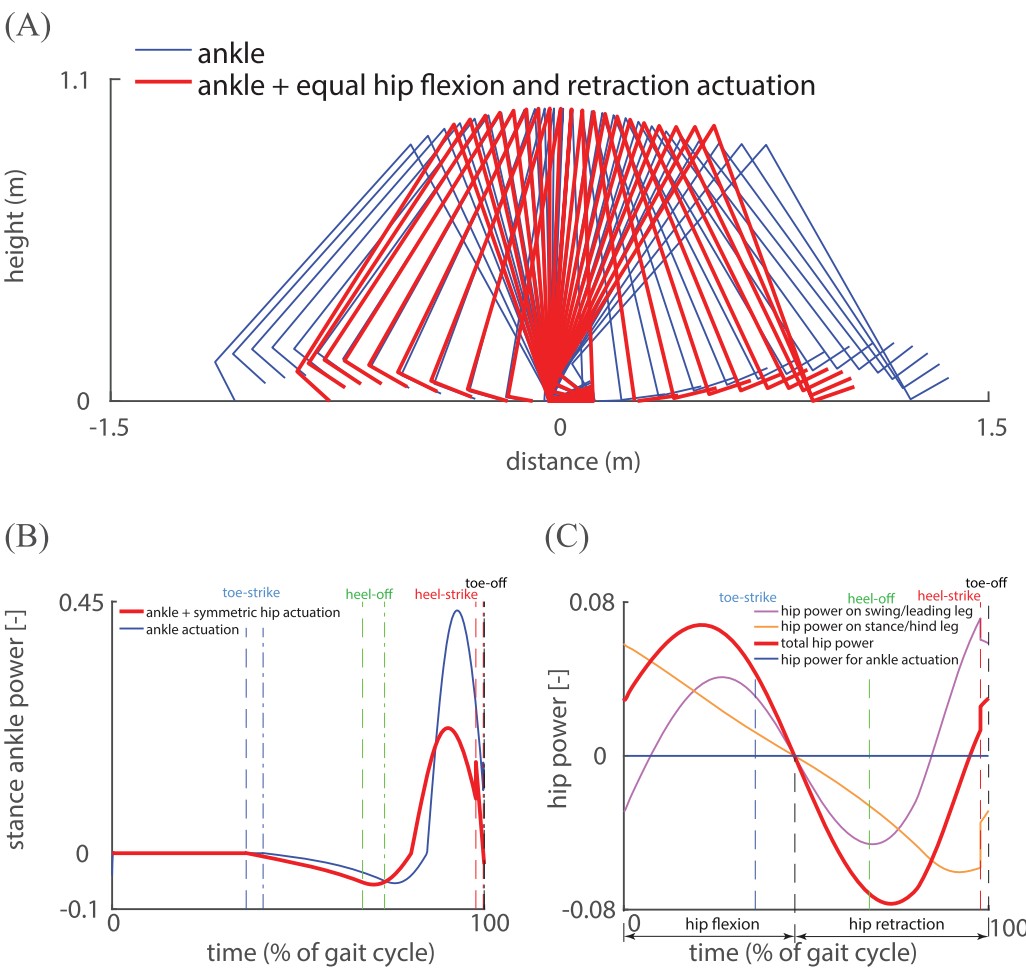

**Figure 3** **Comparison of kinematics and mechanical power for two actuated periodic gaits.** (A) Stick diagram of the periodic gaits for only ankle actuation (blue) and ankle actuation with equal amounts of hip flexion and retraction actuation (red) at speed 0.47. The addition of hip flexion and retraction actuation reduces step length; (B) time-normalized ankle power for ankle-actuated optimal gait (blue) and for a periodic gait with a symmetric hip spring (red). The positive ankle power indicates ankle push-off. Note that toe-off (black vertical dashed line) occurs shortly after heel strike (red vertical dashed line), indicating an almost instantaneous double stance phase; (C) time-normalized hip power for ankle actuation with a hip spring. Hip power on the swing/leading leg (purple) and stance leg/hind leg (orange) is different because of different angular velocities of the two legs. For only ankle actuation, hip power is zero. Note that the peak hip power is much smaller in magnitude than peak ankle power.

We first illustrate the kinematics of the model (blue stick diagram in Fig. 3A) and how mechanical power is generated at the ankle joint over a gait cycle for an optimal only ankle actuation gait. As indicated by the blue curve of ankle power over a gait cycle in Fig. 3B, starting from mid-stance, the ankle performs negative work before ankle reversal, followed by a large burst of positive work during push-off. The MCOT for this gait was 0.41. The ankle push-off work, which compensates for energy losses (here due to both collision loss and ankle negative work), is performed mostly before heel strike. As shown by *Kuo (2002)* and *Ruina, Bertram & Srinivasan (2005)*, such a pre-emptive ankle push-off

strategy decreases the collision loss by changing the direction of the pelvis velocity before heel strike.

Over a gait cycle, negative mechanical work performed at the ankle and hip as well as the collisions need to be compensated by an equal amount of positive mechanical work at the ankle and hip joint. To understand the energetic effect of hip actuation, it is essential to understand both (1) how hip actuation performs mechanical work and (2) how hip actuation influences collision loss. Figure 3A shows a stick diagram and Fig. 3C shows the mechanical power at the hip over a gait cycle for a gait with ankle actuation and equal hip flexion and retraction actuation at the same speed. The MCOT for this gait was 0.50. The first noticeable change in Fig. 3A is that the step length is substantially shorter than the gait with only ankle actuation, leading to a lower collision loss. Figure 3B shows that the addition of symmetric hip actuation reduces the positive ankle power over the gait cycle. Figure 3C illustrates how mechanical power is performed at the hip: during the single stance phase, hip flexion and retraction torques accelerate and decelerate the swing leg, which increases step frequency and decreases step length. During the (short) double stance phase, push-off was aided by the hip extension torques, as indicated by the positive hip power on the leading leg. By mainly actuating the swing leg which has smaller mass than the pelvis, hip actuation influences the swing foot trajectory, step length and collision loss.

As mentioned in the Methods section, metabolic work in our model was obtained by summing all the positive and negative mechanical work performed by the ankle and hip actuators divided by the corresponding efficiencies of performing positive or negative work. MCOT is then computed by dividing metabolic work per step by the step length. In a periodic gait, the total metabolic cost for performing negative work is higher than the metabolic cost directly associated with this negative work. This is because negative mechanical work needs to be compensated by an equal amount of positive mechanical work, which is metabolically costly. As such, only about $\frac{1}{1.2} / \left( \frac{1}{0.25} + \frac{1}{1.2} \right) \approx 17\%$ of the total metabolic cost is associated with actually performing negative work. In general, to understand the energetic effect of adding independent hip flexion and retraction actuation, (1) mechanical energy analysis is required to explain how hip actuation influences ankle/hip positive/negative work and collision loss; (2) metabolic energy analysis is required to explain how these mechanical work components divided by metabolic efficiency of performing positive or negative work and step length together influence the MCOT. These analyses are applied in the following sections.

## Energetics of ankle actuation

Before investigating the effects of hip actuation on MCOT, we first display for different speeds the MCOT and step length for only ankle actuation, which serves as a baseline for the addition of hip actuation. At each speed, eight state parameters and two control parameters (ankle spring stiffness and ankle pulse torque) were optimized to obtain the optimal ankle actuation gait that has the lowest MCOT. Figure 4 shows the MCOT and step length for optimal ankle actuation gaits at different speeds. From Figs. 4A and 4B, it can be seen that both MCOT (0.19–0.51) and step length (0.5–1.4) increase monotonically

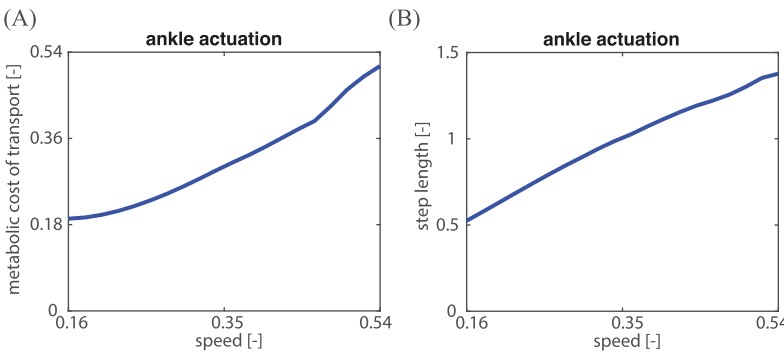

**Figure 4  MCOT and step length over speeds for ankle actuation.** (A) The relations between speed and MCOT for the optimal ankle actuation gait; (B) the relations between speed and step length for the optimal ankle actuation gaits. All values were shown in non-dimensional units.

with speed (0.16–0.54). A monotonic increase of MCOT with speed, and an almost linear relation between speed and step length was also found by *Kuo (2002)*.

## Can MCOT be reduced by adding only hip flexion actuation?

Hip flexion actuation provides direct control over the swing leg, but it is not clear if, and if so how, the addition of only flexion actuation can reduce MCOT. To investigate this, we studied the model's mechanical/metabolic energy at a low, medium and high speed (0.21, 0.38, 0.54 respectively). At each of these three speeds, we added increasing amounts of hip flexion actuation, and for each hip flexion actuation, we searched for the optimal parameters resulting in a periodic gait with the lowest MCOT. While we succeeded in finding periodic gaits for a range of hip flexion actuation, the feasible range was quite small for high speeds, rendering the hip flexion actuation there negligible. Figures 5A and 6A demonstrate the MCOT as a function of hip flexion stiffness for a low and medium speed, and show that adding hip flexion actuation does not lower MCOT. In fact, the gait with the lowest MCOT was a gait with zero flexion actuation (indicated by the blue dots in Figs. 5A and 6A). The stick diagram for zero hip flexion actuation (and thus only ankle actuation) and the highest value of hip flexion actuation (leading to a 16.8% increase in MCOT) at a low walking speed are shown in Fig. 5B. From this figure, it is clear that the hip flexion actuation raised the swing leg, which caused a greater collision loss mainly due to larger (negative) impulsive work in vertical direction, as shown in Fig. 5C (and similarly for a medium speed, see Fig. 6C). The vertical impulsive work is the product of the vertical impact velocity and the vertical impulse. Figures 5D and 6D show that both the vertical impact velocity and vertical impulse increased when adding hip flexion actuation.

An additional reason for an increase in MCOT (Fig. 5A) could be a decrease in step length. However, the step length decreased only slightly at a low speed (see Fig. 5B) compared to the increase in metabolic work per step (see the solid curve in Fig. 5F), and step length even increased at a medium speed (see Fig. 6B). Figures 5E and 5F show the mechanical and metabolic work components for a gait at a low speed. From these two figures it is clear that the increase in hip positive mechanical (and metabolic) work exceeds the reduction in

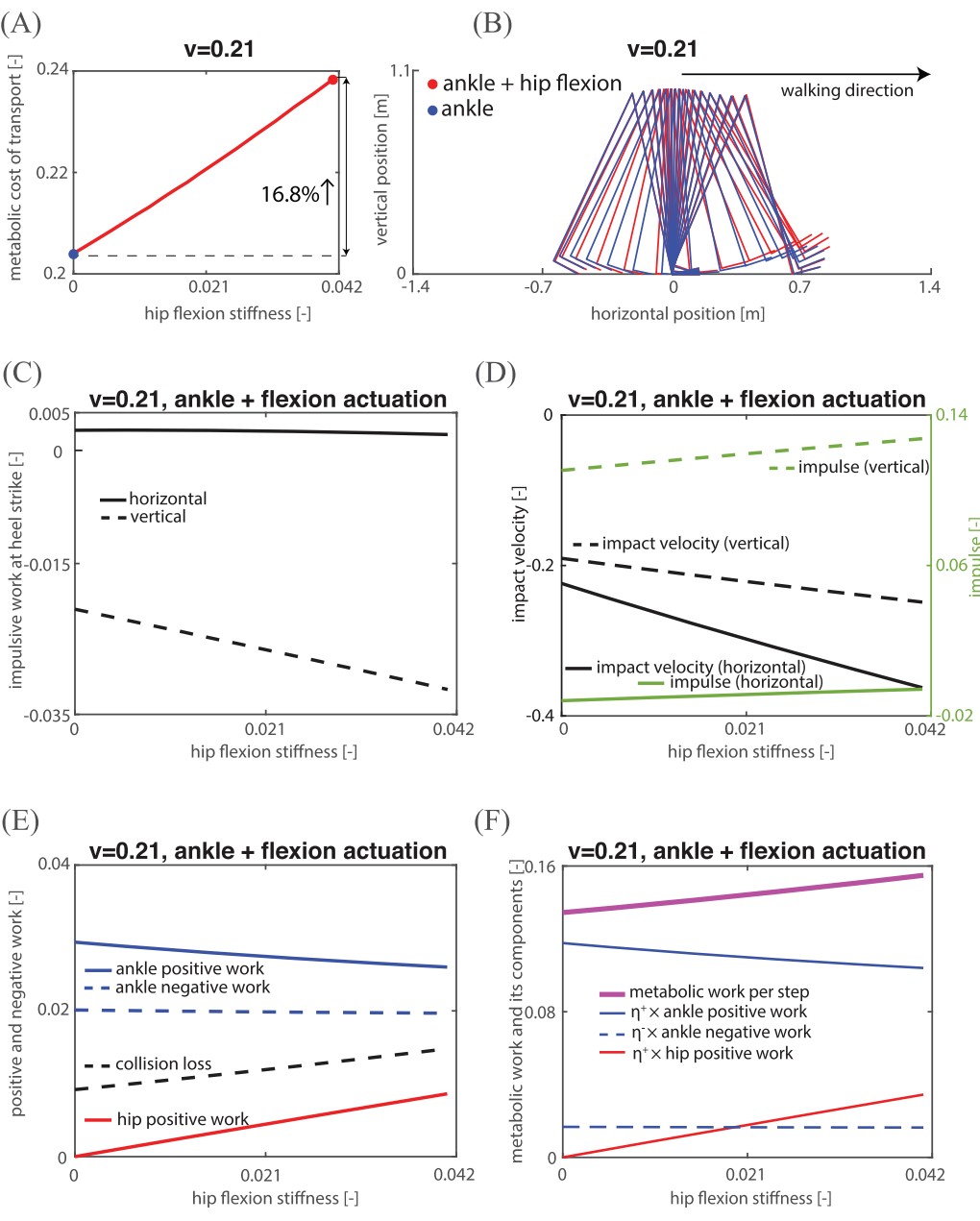

**Figure 5** **Kinematic, mechanical and energetic analysis for a low speed.** (A) The relations between hip flexion stiffness and MCOT. The blue dot represents the ankle actuation gait, and red dot represents the ankle and maximal flexion actuation gait; (B) stick diagram of the two actuation gaits; (C) impulsive work at heel strike in both horizontal and vertical directions; (D) impact velocity and impulse in both horizontal and vertical directions; (E) all positive and negative work components; (F) metabolic work and all its components.

ankle positive mechanical (and metabolic) work, leading to higher total metabolic work per step. To conclude, hip flexion actuation causes an increase in MCOT mainly due to the higher collision loss from larger vertical impact velocity and vertical impulse, which needs larger total mechanical and thus metabolic work at ankle and hip joint to compensate.

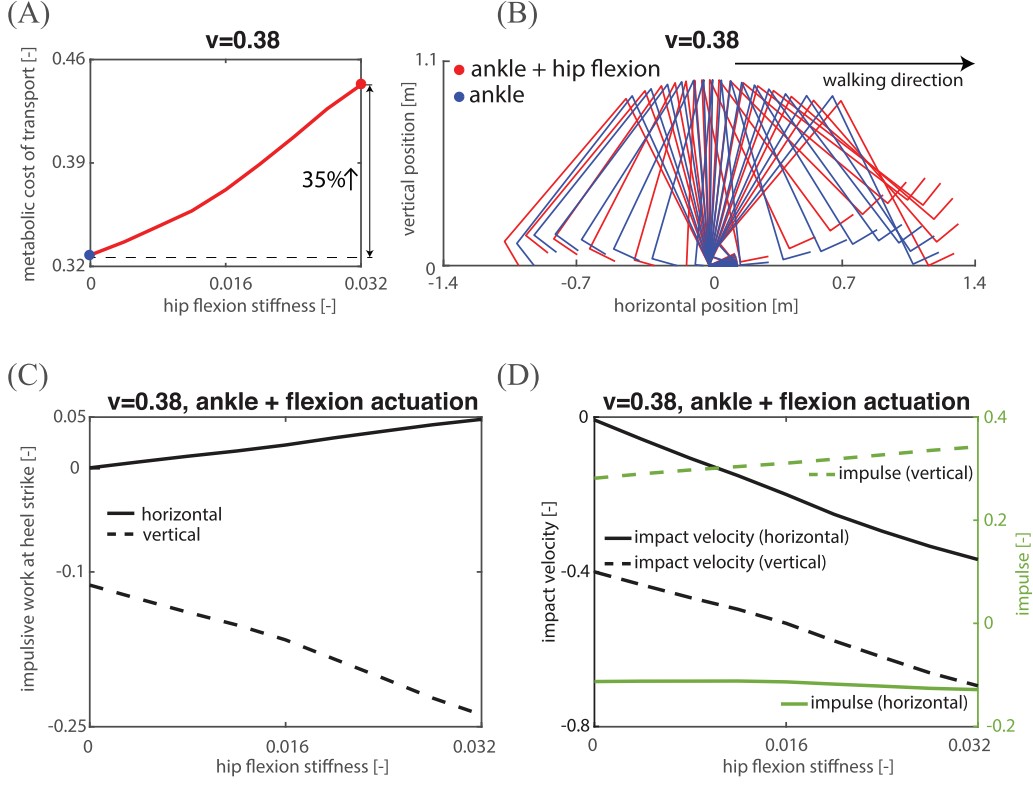

**Figure 6 Kinematic, mechanical and energetic analysis for a medium speed.** (A) The relations between hip flexion stiffness and MCOT; (B) stick diagram of the two actuation gaits; (C) impulsive work at heel strike in both horizontal and vertical directions; (D) impact velocity and impulse in both horizontal and vertical directions.

## Can adding hip retraction actuation in addition to hip flexion actuation reduce MCOT?

As shown above, adding hip flexion actuation resulted in higher MCOT compared to ankle actuation, mostly due to the larger vertical impact velocity and vertical impulse leading to an increase in collision loss. Adding hip retraction actuation could potentially reduce the collision loss by reducing the vertical impact velocity and impulse. However, the reduced collision loss would be at the cost of higher hip negative work, which, as discussed before, requires an equal amount of positive work to compensate. Moreover, hip retraction actuation reduces step length, which leads to higher MCOT. Motivated by these tradeoffs, in this section, we investigated the effect of adding hip retraction actuation for a given hip flexion actuation on MCOT.

We found that only ankle actuation led to the lowest MCOT at both low and high speeds, as discussed later in this section. At medium speeds, the addition of (optimal) hip retraction actuation reduced MCOT compared to any given flexion actuation gait, as can be seen in Fig. 7A. Interestingly, for the zero hip flexion actuation, adding optimal retraction actuation led to a gait with a lower MCOT compared to the only ankle actuation gait (Fig. 7A). To understand why adding only retraction actuation is optimal, we investigated the effects of adding retraction actuation (with no hip flexion actuation) on

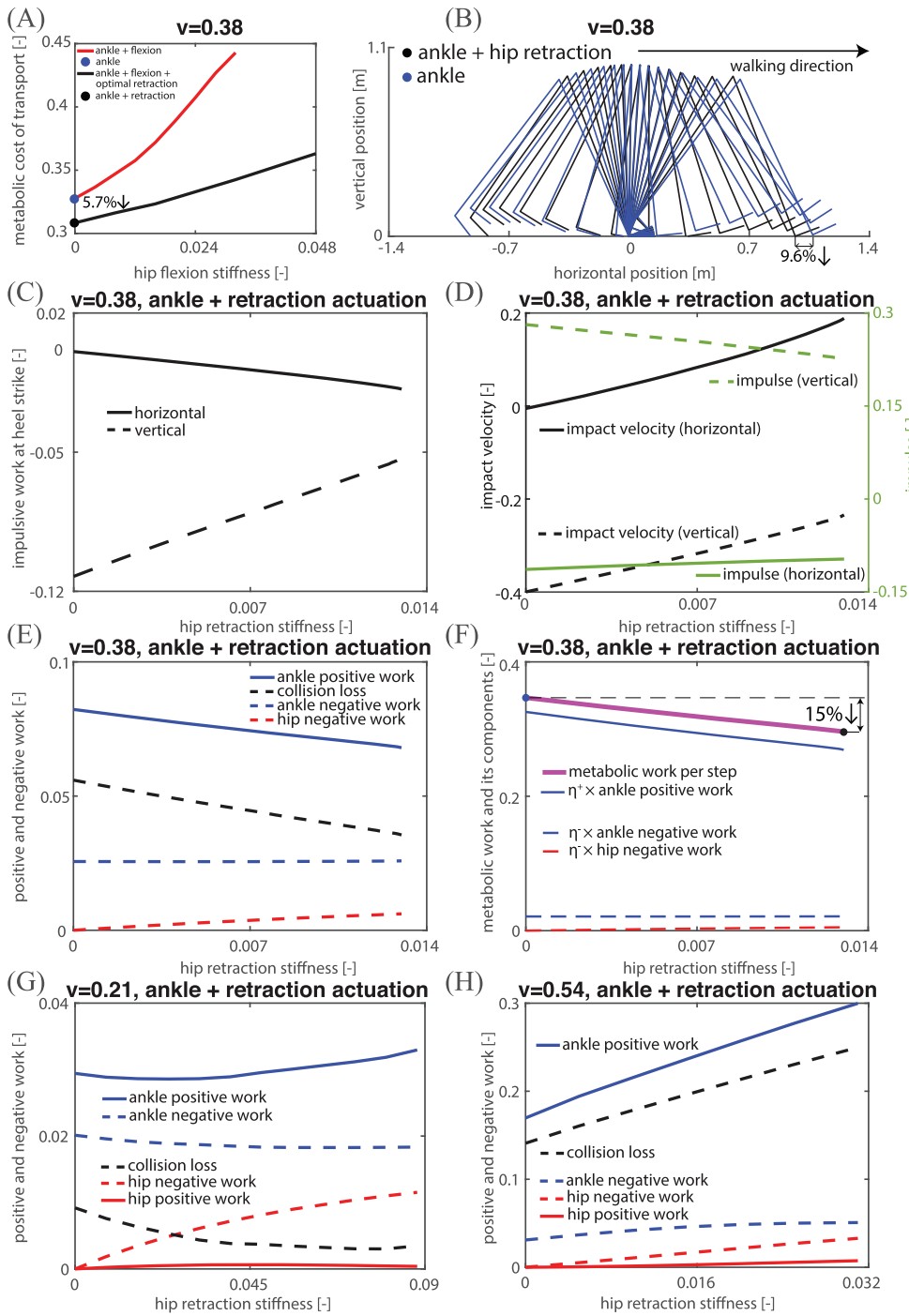

**Figure 7 Kinematic, mechanical and energetic analysis.** (A) The relations between hip flexion stiffness and MCOT for flexion actuation and flexion with optimal retraction actuation at a medium speed; (B) stick diagram of periodic gaits for ankle actuation and retraction actuation; (C) impulsive work at heel strike in both horizontal and vertical directions for retraction actuation at a medium speed; (D) impact velocity and impulse in both horizontal and vertical directions for retraction actuation at a medium speed; (E) all positive and negative work components for retraction actuation at a medium speed; (F) metabolic work and all its components for retraction actuation at a medium speed; (G and H) all positive and negative work components for retraction actuation at a low and high speed respectively.

collision loss, impact velocity and impulse, as illustrated in Figs. 7C and 7D. From these figures, it can be seen that the collision loss decreases with increasing hip retraction actuation. The stick diagram in Fig. 7B provides an intuitive reason why this is so: the swing heel height is closer to the ground for the optimal hip retraction actuation gait, resulting in lower vertical impact velocity (Fig. 7D). Figure 7E shows that increasing retraction actuation reduced collision loss and only slightly increased hip negative work, resulting in overall lower total positive and negative mechanical work. The optimal retraction actuation led to a reduction in metabolic work per step (15%, see Fig. 7F), but also a reduction in step length (9.6%, see Fig. 7B), together resulting in about 5.7% reduction in MCOT compared to only ankle actuation (Fig. 7A).

At low and high speeds, adding hip flexion actuation with optimal retraction actuation led to a higher MCOT. Figure 7G shows that at a low speed, adding hip retraction actuation reduced collision loss. However, the increase in required hip negative work was more than the reduction in collision loss, which resulted in more positive work from the ankle and hip, and thus, higher total metabolic work. Combined with the fact that the addition of only hip retraction actuation led to shorter step length, adding hip retraction actuation led to higher MCOT for these lower speeds. Figure 7H shows that at a higher speed, increasing hip retraction actuation actually led to larger collision loss due to larger vertical impact velocity. As such, both the collision loss and the negative hip work increased with hip retraction actuation, leading to substantially higher total positive work needed and higher metabolic work per step, resulting in higher MCOT.

To conclude: Ankle actuation with some hip retraction actuation was optimal in terms of MCOT at medium speeds (0.32–0.44), with average and maximal reductions of 4.6% and 6% in MCOT compared to only ankle actuation. At all other speeds, only ankle actuation was optimal. These optimal gaits with hip retraction actuation at medium speeds had a low swing heel trajectory above the ground, decreasing the vertical impact velocity, which reduced collision loss.

## DISCUSSION

We evaluated the independent effect of hip flexion and retraction actuation on the MCOT at different speeds in a simple model of human walking. We found that ankle actuation only was optimal at low and high speeds, and only at medium speeds did the addition of hip retraction reduce MCOT (by maximally 6%) compared to ankle actuation.

### Effects of hip actuation on collision loss

The relation between impact velocity, impulse and (ground) impulsive work (Eq. (4)) has been shown analytically in *Font-Llagunes & Kövecses (2009)*; in Supplemental Material B we provide a more intuitive proof. To the best of our knowledge, this relation has not yet been applied to demonstrate the effect of hip actuation on collision loss. We found that adding hip flexion actuation leads to higher collision loss due to both a larger vertical impact velocity and larger vertical impulse. The opposite is true for adding hip retraction actuation. However, reducing the collision loss does not necessarily yield a lower MCOT, because lower impact velocity requires hip retraction actuation performing more negative

work. This comes with an increase in total positive work and metabolic work per step and results in higher MCOT.

## Effects of hip actuation on MCOT

MCOT is determined by metabolic work per step and step length, and metabolic work is computed from ankle and hip mechanical work. To analyze the mechanisms of optimal gaits, it is useful to study the effects of hip actuation on mechanical work components (ankle/hip positive/negative mechanical work, collision loss) and step length, and how these components contribute to the MCOT. Here, adding hip flexion actuation increased the collision loss substantially due to larger vertical impact velocity and vertical impulse, resulting in higher total positive mechanical work, metabolic work per step and MCOT. Moreover, at medium speeds (0.32–0.44), adding hip retraction actuation with zero flexion actuation led to a larger reduction in metabolic work per step (in percentage) relative to the reduction in step length. This resulted in a reduction in MCOT of maximally 6% compared to ankle actuation only. For low and high speeds, such an energetic benefit of hip retraction actuation was absent. Apparently, at low speeds, the collision loss is already small and the reduction in collision does not outweigh the increase in hip negative work, leading to higher total mechanical work (Fig. 7G) and higher MCOT. At high speeds, adding hip retraction actuation causes an increase in collision loss (Fig. 7H) due to higher vertical impact velocity, resulting in higher MCOT.

We investigated the effects of hip flexion and retraction actuation on MCOT in a 'simple' model with the aim to understand the mechanisms underlying these effects. These predictions are based on specific model features like actuation types and based on assumptions of specific metabolic cost functions. Therefore, we elaborate next on how these model details and assumptions influence the predictions we made about optimal gaits and underlying mechanisms.

## Model simplifications

Our model lacked, *e.g.*, knee, trunk, and muscle dynamics, because we sought to highlight fundamental mechanisms underlying the optimal gaits. Here, we justify some of the simplifications we made but also discuss some pitfalls. Our ankle actuation model consisted of an ankle spring and pulse torque, which allowed for adjustments of push-off timing and magnitude, and from which the resulting angle-torque relation is roughly similar to the angle-torque relation in human walking (*Zelik et al., 2014*). The pulse torque was initiated when the angle changed from dorsiflexion to plantarflexion (see Fig. 2A), while in humans, the increase in ankle torque is more gradual (*Shamaei, Sawicki & Dollar, 2013*). This results in gaits with step lengths larger than 1 m at medium and high speeds in our model, because for these gaits the shorter rise time of ankle torque (compared to human ankle torque) can lead to earlier peak angular acceleration of the ankle and thus a larger ankle angle at contralateral heel strike. We modelled spring-like hip actuation, which is different from the burst-like hip torques observed in human walking (*Doke, Donelan & Kuo, 2005*). However, the spring-like hip actuation and burst-like hip actuation can similarly modulate the step frequency (*Kuo, 2002*). Our model was also able to

generate hip extension torques after heel strike (see Fig. 3C), which is an important source of work for push-off in human walking (*Browne & Franz, 2017*; *Umberger, 2010*; *Winter, 1983*). Therefore, the ankle and hip actuation can be considered realistic in their roles in push-off propulsion and swing leg control, and thus capture important characteristics of human walking.

## Metabolic cost simplifications

The metabolic cost of muscle contraction can generally be partitioned into muscle (de-)activation and cross-bridge cycling (*Homsher & Kean, 1978*; *Woledge, Curtin & Homsher, 1985*). There have been several attempts to model the link between muscle mechanics and muscle energetics (*Anderson & Pandy, 2001*; *Bhargava, Pandy & Anderson, 2004*; *Lichtwark & Wilson, 2005*; *Umberger, Gerritsen & Martin, 2003*) or to directly predict muscle energetics from cross-bridge models (*e.g.*, Huxley models; cf. *Huxley, 1957*; *Julian, 1969*; *Lemaire et al., 2016*). The extent to which these models are capable of adequately predicting metabolic cost of muscle contraction and, in addition, of locomotion involving complex musculoskeletal models is still under debate. Several studies have investigated the relative importance of various muscle mechanical factors (*e.g.*, force production, force rate and work; cf. *Doke & Kuo, 2007*; *Kuo, 2001*; *Umberger & Rubenson, 2011*) in predicting metabolic cost, but reported results were not consistent (*Beck et al., 2022*; *van der Zee & Kuo, 2021*), and the predictions were "very sensitive to the metabolic model, muscle model and neural controller" (*Hicks et al., 2015*; *Miller, 2014*). All in all, accurately predicting metabolic cost is far from straightforward. Here, we chose the metabolic work as proxy for overall metabolic cost of walking. The validity of this proxy can be justified by the fact that in both isolated leg swinging (*Doke, Donelan & Kuo, 2005*) and in locomotion (*Riddick & Kuo, 2022*), the joint mechanical power and metabolic power are monotonically related, suggesting that lower metabolic work can be a proxy for lower metabolic cost and higher energy efficiency. Still, we overestimated the metabolic cost at the ankle and hip joints (dimensionless MCOT for humans is 0.23 and for our optimal gait is 0.38 at the same human preferred speed), due to the fact that humans generate burst-like hip torques rather than spring-like hip torques, and due to storage and release of elastic energy at tendons. For instance, the Achilles tendon was shown to passively store and release up to 50% of the total mechanical work involved in a gait cycle of running (*Ker et al., 1987*; *Sasaki & Neptune, 2006*). As a result, our optimization results should be interpreted based on the assumption that all ankle and hip joint torques are actively generated by muscles, with implications to human walking discussed in the next paragraph. The exact differences between the predictions made from our model and predictions that could have been made from other (more complicated) models are beyond the scope of our current work.

## Implications to human walking

We investigated the effect of hip flexion and retraction actuation on the MCOT in a simple model. The energetic effect of varied hip actuation and ankle actuation was previously investigated in modeling (*Kuo, 2002*; *Neptune, Sasaki & Kautz, 2008*) and experimental studies (*Lewis & Ferris, 2008*; *Pieper et al., 2021*; *Teixeira-Salmela et al., 2008*; *Umberger &*

*Martin, 2007*). These studies indicated that strong hip actuation increases MCOT when hip actuation cost is included, and is beneficial when this actuation cost is ignored. In our model, we also found that the MCOT decreased with larger 'free' symmetric hip actuation, see Supplemental Material D. In human walking, hip flexion torques ($\sim$0.4 and $\sim$1 Nm per kg of body mass at slow and fast speed) are larger than hip retraction torques (0 and $\sim$0.3 Nm per kg of body mass at slow and fast speed) during the swing phase (*Winter, 1984*). It seems that the substantial hip flexion torques found at all speeds in humans cannot be explained by our model prediction that any hip flexion actuation is inefficient. The reasons for this disagreement may be that we overestimated the metabolic cost of hip actuation compared to humans, that step lengths in human walking are generally smaller than our model predictions, and that hip flexion torques may serve other objectives such as trunk angular control (*Nott et al., 2010*), trunk stabilization and gait robustness (*Deng, Zhao & Xu, 2017*; *Rummel & Seyfarth, 2010*; *Wisse, Hobbelen & Schwab, 2007*). For instance, *Deng, Zhao & Xu (2017)* found that adding hip torsional springs between the torso and leg is necessary for trunk stabilization in a simple compass walker model. *Hasaneini et al. (2013)* used a telescoping ankle and hip actuated walker model with trunk to optimize the MCOT, and found that ankle push-off and hip actuation contributed equally to the metabolic cost. Note that our model did not include the trunk because our main goal was to study the *energetic* effect of hip actuation, whereas for a walker model with trunk, this energetic effect is dependent on trunk stabilization, making it difficult to interpret the energetic effect. The absence of hip retraction torques at slow speeds in human walking is consistent with our predictions, suggesting that the energy inefficiency of hip retraction actuation at slow speeds is likely to be responsible for its absence in human walking. The considerable magnitude of hip retraction torques in normal speed walking ($\sim$0.1 Nm per kg of body mass; see *Winter, 1984*) also agrees with our predictions that some retraction actuation at medium speeds is energy efficient. At high speeds, humans generate larger hip retraction torques, which disagrees with our prediction that ankle actuation only is optimal.

The reason for this disagreement may be that a higher step frequency requires at least some hip retraction actuation, and that hip retraction torques serve other objectives, such as improving gait robustness, as has been suggested in modeling studies (*Hobbelen & Wisse, 2008*; *Wisse, Atkeson & Kloimwieder, 2005*). Taken together, our study showed at best limited energetic benefits of hip flexion and retraction actuation across speeds, whereas humans in general have larger hip flexion and retraction actuation than our predictions, suggesting that hip actuation in humans is likely to play other roles, such as trunk stabilization and improving gait robustness.

## CONCLUSIONS

We studied the effects of independent hip flexion and retraction actuation on the MCOT at different speeds. Ankle actuation only is optimal at low and high speeds. Adding hip retraction actuation can lead to a modest decrease in MCOT compared to ankle actuation only (maximally 6%) at medium speeds. The mechanisms for this lower MCOT from adding hip retraction actuation are a larger reduction in collision loss than the associated increase in hip negative work, which both require positive mechanical work to

compensate, causing a larger reduction in metabolic work per step than the reduction in step length. Taken together, hip flexion actuation does not appear beneficial because it increases the collision loss due to larger vertical impact velocity and vertical impulse.

### Funding
Sjoerd M. Bruijn and Jian Jin are funded by a VIDI grant no. (016.Vidi.178.014) from the Dutch Organization for Scientific Research (NWO). The funders had no role in study design, data collection and analysis, decision to publish, or preparation of the manuscript.

### Grant Disclosures
The following grant information was disclosed by the authors:
Dutch Organization for Scientific Research (NWO): 016.Vidi.178.014.

### Competing Interests
Jaap H. van Dieën is an Academic Editor for PeerJ.

### Author Contributions
- Jian Jin conceived and designed the experiments, performed the experiments, analyzed the data, prepared figures and/or tables, authored or reviewed drafts of the article, and approved the final draft.
- Dinant Kistemaker conceived and designed the experiments, performed the experiments, analyzed the data, prepared figures and/or tables, authored or reviewed drafts of the article, and approved the final draft.
- Jaap H. van Dieën analyzed the data, authored or reviewed drafts of the article, and approved the final draft.
- Andreas Daffertshofer analyzed the data, authored or reviewed drafts of the article, and approved the final draft.
- Sjoerd M. Bruijn conceived and designed the experiments, performed the experiments, analyzed the data, prepared figures and/or tables, authored or reviewed drafts of the article, and approved the final draft.

### Data Availability
The code and generated data are available at Zenodo: Jin, Jian, Kistemaker, Dinant, van Dieën, Jaap, Daffertshofer, Andreas, & Bruijn, Sjoerd. (2022). Code and data for manuscript: The energetic effect of hip flexion and retraction in walking at different speeds: a modeling study (version of first submission to PeerJ). Zenodo. https://doi.org/10.5281/zenodo.7107403.

### Supplemental Information
Supplemental information for this article can be found online at http://dx.doi.org/10.7717/peerj.14662#supplemental-information.

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
