# Peer review of "The energetic effect of hip flexion and retraction in walking at different speeds: a modeling study"

_PeerJ, doi:10.7717/peerj.14662_

## Round 0.1 · original submission · Minor Revisions

Please address the reviewers' suggestions and resubmit your article.

Reviewer 1 ·

Basic reporting

no comment

Experimental design

no comment

Validity of the findings

no comment

Additional comments

General comments
This manuscript aims at evaluating the independent effect of hip flexion and retraction actuation on the metabolic cost of walking at different speeds in a simple model of human walking. Authors found that ankle actuation only was optimal at low and high speeds, and only at medium speeds the addition of hip retraction reduced the metabolic cost of walking (by maximally 6%) compared with ankle actuation. Authors manage to fulfill properly their aims.

Minor comments
(line 604) Kuo, A. D.
(l558, 572 and 636) Some missing data?
(l110) … free: hip torques…

Reviewer 2 ·

Basic reporting

The manuscript overall is well written and clear, with all source files and helpful supplements an thorough references to the field.
I have made specific suggestions for clarity in the "Additional Comments" section.

Experimental design

The paper is model-based, covering theoretical findings about bipedal locomotion. The question is derived from past investigations and is properly grounded in the literature. It has relevance to human locomotion, and provides a variety of references and interpretation in this domain. Therefore it creates a bridge among physical sciences, mathematical sciences and biological sciences, and health/medical. I believe it fits within the aims and scope of the journal.

The technical standard is good and the technical content, especially including the supplement that includes full derivation, is high.

Validity of the findings

Model source code is provided, allowing replication.
There are no source data, as it is a modeling study.
Graphs of the results are detailed and informative.
Discussion and conclusions are clear.

Additional comments

Specific comments on points in the manuscript:

Abstract: 2nd last sentence "Both terms require positive mechanical work to compensate, yielding a larger reduction in metabolic work per step than the reduction in step length.": This reads oddly, as though positive mechanical work yields a reduction in metabolic cost. Consider rephrasing with fewer positive/negative reversals if possible, for clarity.

153: please label mtot earlier in the line where "total mass" is introduced.

159-60: Please state "proximal to distal relative to the ground" (typically these mean relative to the trunk or pelvis)
160-62: Note that this sentence lists COM location as "distance from the proximal end", which if using the same definition of distal and proximal as the sentence above, would cause the COM of the legs to be asymmetrical (one high and one low). These two sentences need to be reworked for clarity.

194-5: damper added to ankle of the leading foot after heel strike: did this continue through stance phase, or just until foot flat? (maybe say "after heel strike until _____")

199: suggest a new paragraph for Hip actuation rather than combined with the end of the ankle actuation section.

Fig 2a: It's not clear to me why there are two points of the curve marked "toe-off" and why "heel strike" is marked at the end rather than the beginning of the cycle. What is the loopy part at the very beginning (bottom center?) - it seems it should start with heel strike (?). And is the part currently called heel-strike actually "opposite heel strike"? Please make a few adjustments for clarity.

222-5: The Methods refer to results Sections 4.2-4.4 but these are not numbered below. Please number or revise the reference.

317: This section heading ("Hip flexion actuation is not energetically beneficial") seems to suggest that it is never energetically beneficial; but the first sentence makes clear this refers to cases with ONLY hip flexion actuation. Consider revising the section heading.

Fig 5d: The "Impulse" axis could benefit from a tick mark at 0.

484-485, 503: units "Nm per body mass" aren't quite clear - does this mean "Nm per kg of body mass"?

---

## Round 0.2 · Minor Revisions

Please, address the remaining issues within the manuscript according to the reviewer 2:

The changes have been made to address my comments. The manuscript is clearer in most of the spots that had trouble before, and makes that much more sense now.

One remaining note: lines 112-114 still seem to use a strange definition of "proximal" to mean, roughly, "proximal to the extremity". This remains at odds with the standard anatomical use of that term, which means, roughly, "proximal to the heart". Fig 1 is enough to make it clear, so it's okay; but the language could still be updated. To make a suggestion: perhaps rephrase 110-114 something like this:

"... The subscripts refer to the segments NUMBERED ALONG A KINEMATIC CHAIN STARTING WITH the toe of the stance foot (i.e., 1 = stance foot, 2 = stance leg, 3 = swing leg, 4 = swing foot). Every segment configuration was determined by four parameters: center-of-mass m_i, length l_i, distance from the DISTAL end to the center-of-mass d_i, and moment of inertia relative to the center-of-mass j_i. ..."

This or similar phrasing would eliminate "proximal" from the definition of the kinematic chain and retain it in the traditional anatomical usage.

Reviewer 1 ·

Basic reporting

no comment

Experimental design

no comment

Validity of the findings

no comment

Additional comments

General comments
I do not have any further particular concerns to express about the manuscript. Authors addressed sufficiently all points raised by the two reviewers.

Reviewer 2 ·

Basic reporting

Very good, no change in assessment.

Experimental design

Very good, no change in assessment.

Validity of the findings

Very good, no change in assessment.

Additional comments

The changes have been made to address my comments. The manuscript is clearer in most of the spots that had trouble before, and makes that much more sense now.

One remaining note: lines 112-114 still seem to use a strange definition of "proximal" to mean, roughly, "proximal to the extremity". This remains at odds with the standard anatomical use of that term, which means, roughly, "proximal to the heart". Fig 1 is enough to make it clear, so it's okay; but the language could still be updated. To make a suggestion: perhaps rephrase 110-114 something like this:

"... The subscripts refer to the segments NUMBERED ALONG A KINEMATIC CHAIN STARTING WITH the toe of the stance foot (i.e., 1 = stance foot, 2 = stance leg, 3 = swing leg, 4 = swing foot). Every segment configuration was determined by four parameters: center-of-mass m_i, length l_i, distance from the DISTAL end to the center-of-mass d_i, and moment of inertia relative to the center-of-mass j_i. ..."

This or similar phrasing would eliminate "proximal" from the definition of the kinematic chain and retain it in the traditional anatomical usage.

---

## Round 0.3 · accepted · Accept

The authors have addressed all of the reviewers' comments. The manuscript is ready for publication.

Reviewer 1 ·

Basic reporting

General comments
I confirm not to have any further particular concerns to express about the manuscript. Authors addressed sufficiently all points raised by the two reviewers.

Experimental design

No comment.

Validity of the findings

No comment.

Additional comments

No comment.

Reviewer 2 ·

Basic reporting

no comment

Experimental design

no comment

Validity of the findings

no comment

Additional comments

The authors have addressed all my concerns adequately.